# Biological network growth in complex environments: A computational framework

**Torsten Johann Paul, Philip Kollmannsberger**[ID]*

Center for Computational and Theoretical Biology, University of Würzburg, Campus Hubland Nord 32, Würzburg, Germany

* philip.kollmannsberger@uni-wuerzburg.de

## Abstract

Spatial biological networks are abundant on all scales of life, from single cells to ecosystems, and perform various important functions including signal transmission and nutrient transport. These biological functions depend on the architecture of the network, which emerges as the result of a dynamic, feedback-driven developmental process. While cell behavior during growth can be genetically encoded, the resulting network structure depends on spatial constraints and tissue architecture. Since network growth is often difficult to observe experimentally, computer simulations can help to understand how local cell behavior determines the resulting network architecture. We present here a computational framework based on directional statistics to model network formation in space and time under arbitrary spatial constraints. Growth is described as a biased correlated random walk where direction and branching depend on the local environmental conditions and constraints, which are presented as 3D multilayer grid. To demonstrate the application of our tool, we perform growth simulations of a dense network between cells and compare the results to experimental data from osteocyte networks in bone. Our generic framework might help to better understand how network patterns depend on spatial constraints, or to identify the biological cause of deviations from healthy network function.

**Data Availability Statement:** All code and data to reproduce the results are available from https://github.com/CIA-CCTB/pythrahyper_net.

## Author summary

We present a novel modeling approach and computational implementation to better understand the development of spatial biological networks under the influence of external signals. Our tool allows us to study the relationship between local biological growth parameters and the emerging macroscopic network function using simulations. This computational approach can generate plausible network graphs that take local feedback into account and provide a basis for comparative studies using graph-based methods.

## Introduction

Complex biological networks such as the neural connectome are striking examples of large-scale functional structures arising from a locally controlled growth process [1, 2]. The resulting

**Funding:** This publication was supported by the Open Access Publication Fund of the University of Wuerzburg.

**Competing interests:** The authors have declared that no competing interests exist.

network architecture is not only genetically determined, but depends on biological and physical interactions with the microenvironment during the growth process [3, 4]. In this context, evolution has shaped diverse spatial networks on all length scales, from the cytoskeletal network in cells [5], to multicellular networks such as the vascular system [6] or the osteocyte lacuno-canalicular network [7], to macroscopic networks of slime molds [8], mycelia [9] and plants [10]. Sophisticated imaging techniques together with large-scale automated analysis provide increasingly detailed views of the architecture of biological networks, revealing e.g. how neurons are wired together in the brain [11]. After extracting the topological connectivity from such image data, quantitative methods from the physics of complex networks can be applied to compare different types of networks and to uncover common organizational principles [12–15]. To understand the functional role of networks in the context of evolution, however, it is not sufficient to characterize static network structure [1]: we need to be able to infer the dynamics of the underlying biological growth process that defines this structure.

This poses a major challenge, since the local dynamics of the growth process is in most cases not experimentally accessible, except for simple model systems. Here, computer simulations provide a solution: they allow to connect the observed patterns to the underlying process by performing *in-silico* experiments. Different hypotheses can be tested by systematically varying the local growth rules in the simulation and analyzing the resulting network patterns. Such a computational approach also helps to overcome an important limitation of network physics with respect to spatial networks: the properties of complex networks are often calculated with respect to canonical random graphs such as the Erdös-Renyi [16] or Watts-Strogatz model [17]. These models are defined by the topological structure in an adjacency matrix, but usually neglect spatial constraints [18, 19]. A generic model of network growth under spatial constraints would thus address two important problems: enabling a meaningful quantification of spatial network patterns, and linking the observed patterns to the underlying biological growth process.

The first computer simulation of spatial network development, published in 1967 [20], already included the role of active and repulsive cues. Since then, several more advanced approaches to model spatial network growth have been developed. One type of model implements network growth by formation of edges between pre-existing nodes in space [21, 22] and was e.g. used to develop large-scale models of branching neurons [23] or leaf vascular networks [24]. Another type of approach is based on growth processes emanating from predefined seed points with branching and merging and has been used to model neural network development [25, 26], fibrous materials [27] or the development of branching organ structures [28]. Most of these existing models are targeted towards a specific domain such as neural networks and synapse formation, or include only simplified interactions with the environment. The current state of modeling spatial network growth was recently summarized in [1], arguing that there is "a crucial lack of theoretical models".

The aim of this work is to develop a generic framework for biological network development in space and time under arbitrary spatial constraints. We propose a probabilistic agent-based model to describe individual growth processes as biased, correlated random motion with rules for branching, bifurcation, merging and termination. We apply mathematical concepts from directional statistics to describe the influence of external cues on the direction of individual growth processes without restricting the model by making too specific assumptions about these cues. Structural and geometric factors are obtained directly from real image data rather than formulated explicitly. The model is implemented as a computational framework that allows us to perform simulations for diverse types of biological networks on all scales, and to monitor the evolving spatial structure and connectivity.

## Results

### Growth process

The elementary process of biological network formation is the outgrowth of new edges in space from existing nodes, such as cells. Each individual growth agent, for example a neuronal growth cone, randomly explores space and senses soluble signals as well as physical cues and obstacles. This process can be formulated as a biased correlated random walk [29]: the movement of edges is *biased* by attractive or repulsive signals and structural cues, and *correlated* due to directional persistence of edge growth. Edges turn into trees if branching and/or bifurcations occur, such as in growing neurons, osteocytes or sprouting angiogenesis. Finally, if edges can join other edges and form new nodes (e.g. synapses), a connected network emerges. The elements of this discretized, agent-based formalization of spatial network development are summarized in Fig 1. In the following, we describe our mathematical approach for modeling the growth process.

In general, such a stochastic motion under the influence of external forces is described by a partial differential equation (PDE), the Fokker-Planck equation [30] corresponding to the Langevin equation [31]:

$$\frac{\partial p(\vec{x}, t)}{\delta t} = -\nabla(\vec{u}p(\vec{x}, t)) + \nabla(\mathbf{D}\nabla p(\vec{x}, t)) \tag{1}$$

Eq 1 is a drift-diffusion equation of motion where the $k$-dimensional vector $\vec{u}(\vec{x}, t)$ is a drift, and $\mathbf{D}$ the anisotropic and diagonal, $k \times k$- dimensional diffusion tensor. This equation describes the time development of a probability density function $p(\vec{x}, t)$ of diffusing particles under the influence of a drift $u(\vec{x}, t)$ and diffusion $\mathbf{D}(\vec{x}, t)$. If drift and diffusion are both constant in time and space, and if the initial condition $p(\vec{x}, 0) = \delta(\vec{x} - \vec{x}_0)$ is the $k$-dimensional

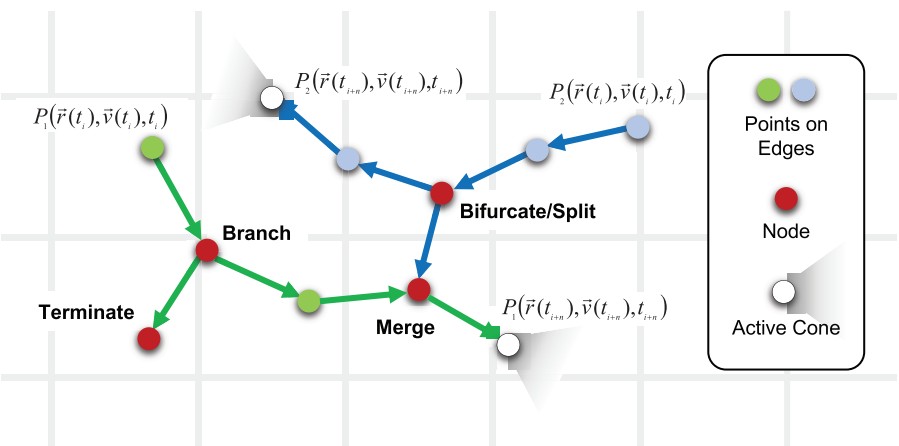

**Fig 1. Discrete local events during spatial network development.** Two growth agents (green and blue, e.g. growth cones, filopodia, cells) start at time $t_i$ and randomly explore their environment in a biased correlated random walk. The environment is represented as a discrete grid (grey lines), while the network grows in continuous space. Blue and green dots correspond to subsequent positions of the growth cones. Cones can branch off new daughter cones, bifurcate/split in two new cones, merge with existing edges, or terminate. These events leads to formation of new nodes in the network, shown as red dots. Over time, the individual processes form a connected network of nodes and edges in space.

Dirac delta function, then Eq 1 solves to

$$p(\vec{x}, t) = \frac{1}{\sqrt{(4\pi)^k |\mathbf{D}t|}} \exp\left(-\frac{1}{2} \sum_{i=0}^{k} \frac{(x_i - x_{0,i} - u_i t)^2}{2 D_i t}\right) \tag{2}$$

Eq 2 describes the time development of a random movement for constant drift and isotropic diffusion starting at initial position $\vec{x}_0$. In a complex environment, however, both drift and diffusion can depend on time and position, and there is no simple analytic solution for the time development. In this case, a numerical approach can be taken by sampling individual movement traces as non-Markovian correlated processes [32]. The drift-diffusion movement is then discretized into a stepwise process where for each growth step, a new direction is drawn from a probability density function (PDF). For constant drift and diffusion (Eq 2), the corresponding PDF, or growth kernel, is a Multivariate Gaussian distribution (MVG) with mean vector $\vec{\mu} = \vec{u}t$, covariance $\Sigma = \mathbf{D}\,t$ and determinant $|\mathbf{D}t| = \prod_{i=1}^{k} 2 D_i t$. In the general case, assuming that the spatio-temporal change for $\mu(\vec{x}, t)$ and $\Sigma(\vec{x}, t)$ is small between two consecutive time steps, every discrete growth step has a different time- and position dependent PDF. The fundamental solutions Eq 2 of the PDE (Eq 1), or Eq 3 for non-diagonal covariance, can thus be used as random kernels in each step from which the subsequent growth directions are drawn. Their $\vec{\mu}$ and $\Sigma$ can be expanded so that they encode for all spatial signals and constraints. A more detailed derivation of this approach starting from Eq 1 is provided as supporting information in S1 Text.

## Substrate and signaling cues

The individual PDFs that describe all the different internal and external constraints and cues that act on an edge at position $\vec{x}_p$ are formulated using the Multivariate Gaussian Distribution (MGD) as kernel:

$$MGD(\vec{x}; \vec{\mu}, \Sigma) = \frac{1}{\sqrt{(2\pi)^k |\Sigma|}} \exp\left[-\frac{1}{2}(\vec{x} - \vec{\mu})^T \Sigma^{-1} (\vec{x} - \vec{\mu})\right] \tag{3}$$

The MGD is a probability density function from which random vectors $\vec{X} \sim N(\vec{\mu}, \Sigma)$ of dimension $k$ are drawn. Such a vector is shifted by the mean vector $\vec{\mu}$ and anisotropically distributed. The anisotropic distribution of directions is described by the metric of the $k \times k$ covariance matrix $\Sigma$ with determinant $|\Sigma|$ and for its eigendecomposition as real symmetric matrix $\Sigma_{diag} = \mathbf{A}^{-T} \Sigma \mathbf{A}$ with eigenvectors $\mathbf{A} = (\vec{v}_1 \ldots \vec{v}_k)$ and eigenvalues $\Sigma_{diag} = diag(\lambda_1 < \cdots < \lambda_k)$.

Depending on the edge's current position $\vec{x}_p$, the mean vector $\vec{\mu}$ points to the most likely direction the edge will be moved in the subsequent step, while the covariance $\Sigma$ reflects the impact of the local structure onto its movement in the complex environment.

Following the mathematical rules for distributions, the PDF of a random vector that is the sum of independent random vectors is the convolution of the PDFs of the individual random vectors:

$$\text{PDF}_{conv} = \text{PDF}_0 * \ldots * \text{PDF}_{n-1} * \text{PDF}_n \tag{4}$$

All local, external signals and structural properties that are described by individual PDFs in the form of a MGD can thus be combined into a single PDF that still has the form of a MGD [33]. As a result, a new growth direction that includes the contribution of all cues and

**Table 1. Mean vector—Covariance pairs.**

| External Cue | $\vec{\mu}$ | $\Sigma$ |
|---|---|---|
| Drift | $\vec{D}(\vec{p}(x,y,z))$ | $\sigma_{x,y,z} \to 0$ |
| Signaling molecules | $\pm \sum_m^n \frac{(\vec{p}_{m,i}^s - \vec{p}_{m,i-1}^s)}{n}$ | $\sum_m^n \Sigma_m$ |
| **Internal Cue** | $\vec{\mu}$ | $\Sigma$ |
| RW - Random Walk | $\vec{0}$ | diagonal, $\sigma_{x,y,z} > 0$ |
| P - Persistence | $\sum_{i=m}^n \vec{p}_i$ | isotropic, diagonal, $\sigma_{x,y,z} > 0$ |
| **Structural Cue** | $\vec{\mu}$ | $\Sigma$ |
| Structure tensors | $\vec{\nabla} p(x,y,z)$ | $\Sigma = \alpha R^T \Sigma_{diag}^{dual} R$ $\Sigma_{diag} = R\Sigma_{i,j}R^T$ $\Sigma_{i,j} = \delta_i p \delta_j p$ $i,j \in (x,y,z)$ |

Table 1: Examples how external, internal and structural cues interact with a growing edge at position $\vec{p}$ in 3D. The physical properties are interpreted as multivariate Gaussian distributions with a mean vector $\vec{\mu}$ and covariance $\Sigma$. All signals are combined into a single PDF by convolution. We distinguish between external signals such as drift and signaling cues, internal properties of an edge such as its random motion and stiffness, and the structural properties of the surroundings. A more detailed version is provided in S2 Table.

constrains can be drawn from this combined MGD with mean vector

$$\vec{\mu}_{\text{conv}} = \sum_{i=0}^n \vec{\mu}_i \tag{5}$$

and covariance

$$\Sigma_{\text{conv}} = \sum_{i=0}^n \Sigma_i \tag{6}$$

These parameter pairs $(\vec{\mu}_i, \Sigma_i)$ can represent different external cues, the geometric constraints of the substrate, as well as edge-specific properties. In Table 1, we give some examples for external and internal cues and the structure of the environment that can be represented by such "Mean Vector—Covariance" pairs.

Typical examples of external cues are drift vector fields and ensembles of signaling molecules. For the case of a constant external vector field, the mean vector points towards the local field orientation, while its length scales with the field's strength. Its corresponding covariance is symmetric, proportional to the unit matrix $\mathbb{I}$, and the entries tend to zero ($\sigma_{1,\ldots,i} \to 0$). The MGD is then a $k$-dimensional delta distribution that biases the growing edge in the direction of the vector field.

Internal signals describe the properties of the growing edge, reflecting its characteristic random motion and its stiffness or persistence. The physical interaction with the substrate is encoded in the structure tensor and the local derivatives (gradients) of the surroundings. The structure tensor takes the place of the diffusive tensor in the MGD, while the mean vector is the local gradient $\vec{p}(x,y,z)$. This concept, which is widely used in image analysis (e.g. anisotropic diffusion) [34–36], allows us to include arbitrary structural information by representing it as an image volume and calculating the corresponding structure tensor.

The combination of the different PDFs through convolution (Eq 4) results in a single MGD from which the subsequent growth direction is drawn. This combined PDF distributes

random vectors around a mean vector with a length that is the sum of all mean vectors of the individual signals. The shape of this shifted multivariate normal is expressed by its covariance.

## Step length

The Multivariate Gaussian Distribution in Eq 3 covers the full $\mathbb{R}^3$, but sampling the entire space to draw the next growth direction would not only be prohibitively slow, but would also generate unphysical step size distributions. This problem can be solved by integrating over the radius to generate a sampling of the PDF from a subset of $\mathbb{R}^3$ (the unit sphere) while retaining the full spatial information contained in its covariance and mean. An additional advantage is that growth direction and step length are separated into two probability distributions which can then be treated independently.

Distributions on the unit sphere $\mathbb{S}^{k-1}$ can be obtained by embedding $\mathbb{S}^{k-1}$ in $\mathbb{R}^k$. For a random vector $\vec{x} \in \mathbb{R}^k$, this could be its conditional distribution on $\|\vec{x}\| = 1$, or its angular projection $\vec{y} = \vec{x}/\|\vec{x}\|$. For the family of Fisher-Bingham distributions [37], the Bingham distribution [38] is the conditional distribution, while the Kent distribution is the projection of the general Fisher-Bingham distribution [39]. Although these distributions are well studied and are commonly used for data fitting and directional statistics [40], the Kent distribution is not suitable for simulations, as it requires a rejection method [41]. Instead, the MGD can be transformed to spherical coordinates, which allows us to calculate a spherical marginal distribution $\text{MGD}_{\text{angular}}$ by integrating over the radial component to obtain a projection of the MGD. Starting from Eq 3, the $k$-dimensional coordinate basis is transformed to $\vec{x} = r\vec{y}$ with $r = \|\vec{x}\|, \vec{y} = \vec{x}/\|\vec{x}\|$ and $d\vec{x} = r^{k-1}dr d\vec{y}$. The transformation and integration of Eq 3 results in the following angular MGD:

$$MGD_{\text{angular}}(\vec{y}; \vec{\mu}, \boldsymbol{\Sigma}) = \int_0^{-\infty} MGD(r, \vec{y}; \vec{\mu}, \boldsymbol{\Sigma}) r^{k-1} dr \tag{7}$$

Carrying out the integration yields the general angular Gaussian distribution (GAGD), introduced and explicitly calculated for dimensions $k$ = 1, 2, 3 in [42], which is the angular marginal distribution to the MGD that projects the directional probability information onto the $\mathbb{S}^{k-1}$-sphere:

$$
\begin{aligned}
MGD_{angular}(\vec{y}; \vec{\mu}, \boldsymbol{\Sigma}) \quad &= \frac{1}{\sqrt{(2\pi)^{(k-1)}|\boldsymbol{\Sigma}|}} \frac{1}{\sqrt{\vec{y}^T \boldsymbol{\Sigma}^{-1} \vec{y}}} \\
&\times \exp\left[\frac{1}{2}\left(\frac{(\vec{y}^T \boldsymbol{\Sigma}^{-1}\vec{\mu})^2}{\vec{y}^T \boldsymbol{\Sigma}^{-1}\vec{y}} - \vec{\mu}^T \boldsymbol{\Sigma}^{-1}\vec{\mu}\right)\right] \\
&\times M_{k-1}\left(\frac{\vec{y}^T \boldsymbol{\Sigma}^{-1}\vec{\mu}}{\sqrt{\vec{y}^T \boldsymbol{\Sigma}^{-1}\vec{y}}}\right)
\end{aligned} \tag{8}
$$

The last term $M_{k-1}\left(\frac{\vec{y}^T \boldsymbol{\Sigma}^{-1}\vec{\mu}}{\sqrt{\vec{y}^T \boldsymbol{\Sigma}^{-1}\vec{y}}}\right)$ of Eq 8 is a function

$$M_{k-1}(\alpha) = \int_{u=0}^{\infty} u^{k-1} \frac{1}{\sqrt{2\pi}} \exp\left(-\frac{(u-\alpha)^2}{2}\right) du \tag{9}$$

For the 3-dimensional case $k = 3$, $\vec{y}^T = (\sin\theta\cos\phi,\ \sin\theta\sin\phi,\ \cos\theta)$, this results in

$$M_2(\alpha) \quad = (1 + \alpha^2)\Phi(\alpha) + \alpha\phi(\alpha)$$

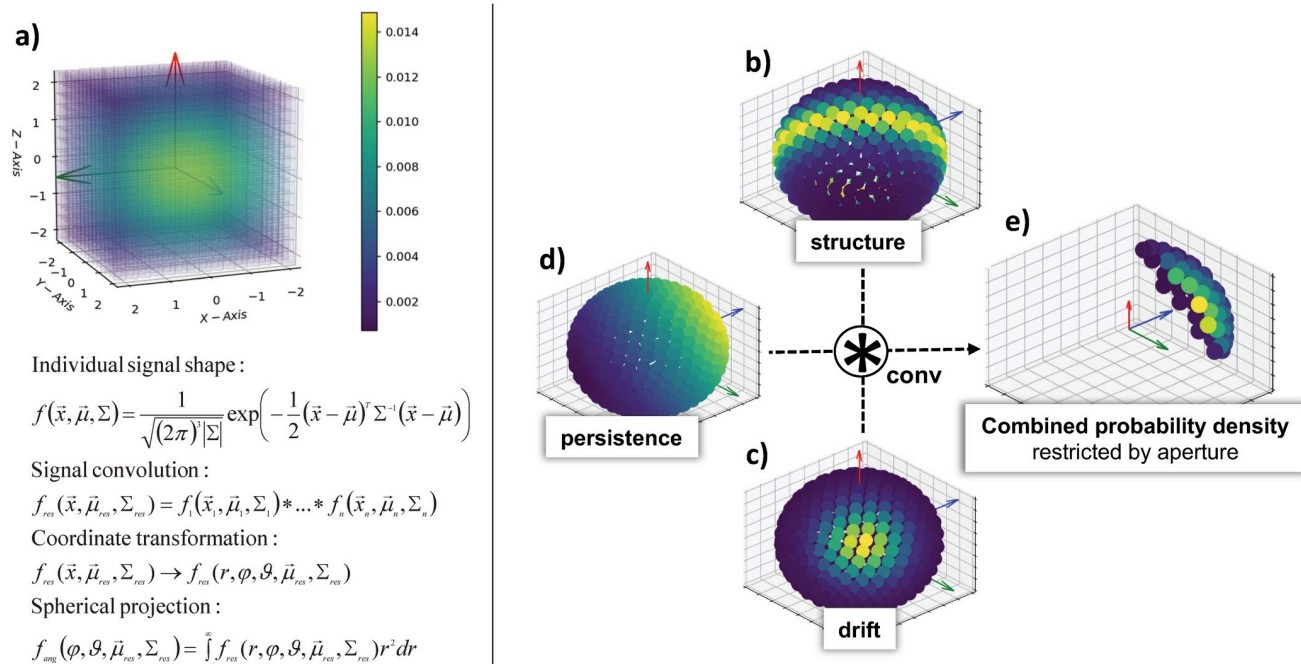

**Fig 2. Probabilistic framework to determine the next growth direction.** Probability distributions are modeled as multivariate Gaussian distributions (MGD), shown in a), with mean and covariance determined from the discrete simulation grid that describes structural and soluble cues. Individual MGDs are combined by convolution, transformed to spherical coordinates, and projected onto the unit sphere (bottom). Examples shown on the right hand side include structural guidance along a plane (e.g. a tissue boundary, b), a unidirectional drift towards the viewer (e.g. a growth factor gradient, c), and persistence due to memory of past growth directions (e.g. bending stiffness, d). The three probability distributions are merged by convolution, restricted by the aperture of the growth cone, and sampled on the corresponding segment of the sphere (e). The next growth direction is drawn from this combined, restricted distribution.

where $\phi(.)$ is the standard normal PDF and $\Phi(.)$ the corresponding CDF, the cumulative distribution function. For additional detail, please see S1 Text.

This definition separates the radial distribution from the angular distribution. While we use a constant step length in our simulations, this separation would also allow for variable step length PDFs without losing any information of the spatial distribution of external cues with large mean vectors $\vec{\mu}$. In Fig 2, we show three distributions on the unit sphere $\mathbb{S}^{k-1} \in \mathbb{R}^k$ for three dimensions with different mean vector $\vec{\mu}$ and covariance $\Sigma$ as examples for shifted isotropic signals (persistence), highly anisotropic covariances due to spatial constraints such as edges and walls, and peaked distributions for guidance cues towards specific directions. The structural information is projected onto the unit sphere while retaining the full information about $\Sigma$ and $\mu$, as shown in Eq 8—e.g., a bimodal MGD still retains a bimodal marginal MGD$_\text{angular}$ due to its covariance.

After motivating the mathematical kernel for the growth process and the interaction with the substrate, we now introduce the rules that enable the growing edges to bifurcate, branch off daughter processes, to form new connections (e.g. synapses) with other edges, and to be persistent and locally self-avoiding.

## Persistence

During edge outgrowth, the finite bending stiffness of edges causes the growth direction to be correlated to the previous directions, resulting in persistence of the movement. In absence of a

drift vector $\vec{\mu}$, and since the eigenvectors of the covariance matrix $\Sigma$ are independent, the random growth process is a Wiener and Markov Process, and still remains a general interpretation of a Wiener process for $\|\vec{\mu}\| > 0$. Persistence of the growth direction is equivalent to introducing a drift that is the sum of the last $t$ growth directions $\vec{\mu}_P = \sum_i^t \vec{\mu}_i$. This results in an effective bending stiffness of the growing edges, and its persistence length is expressed by the Kuhn length [43],

$$P_{\text{Kuhn},t} = \lim_{\delta L \to \infty} \delta L \sum_{j=i+1}^{t} \langle \cos\theta_{i,j} \rangle, \qquad (10)$$

where $\delta l$ is the length between two steps and $\delta l \langle \cos\theta_{i,j} \rangle$ is the projected length at $i$ towards $j$ (Fig 1). This directional correlation does however not imply self-avoidance within the persistence length scale, as there still is a non-zero probability for an edge to loop back and collide with itself within the persistence length.

## Self-avoidance

Local self-avoidance is the phenomenon that growing edges in a biological network, e.g. outgrowing neurites or branching vessels, do not loop back onto themselves [44]. In our growth model, local self-avoidance is achieved by restricting possible growth directions to a solid angle $\Omega$ around $\vec{\mu}_{\text{p}}$. The aperture described by $\Omega$ is equivalent to the maximum allowed curvature of a growing edge. This means that the heuristic growth process is no longer a Markov process [45], as it loses its reversibility, but now becomes self avoiding within the persistence length. Fig 2 shows the effect of persistence on the angular distribution $MGD_{\text{angular}}$ and how the aperture is restricted by $\Omega$.

## Connectivity

New nodes in a growing network form by branching or bifurcation of growing edges, or when a growing edge merges with another edge. We distinguish between "branching", the branching-off of a new edge from a growing edge where the parent edge maintains its growth direction, and "bifurcation", the splitting of a growing edge into two daughter edges growing in two different directions. Both can occur at every growth step with a probability that depends on edge properties (e.g. age) as well as external cues. Whenever an edge is within a certain distance to another edge, it can merge and form a new node, again with a probability that can depend on edge properties and external signals. Finally, growing edges can terminate and form an end-node, again with a probability that depends on external signals, such as obstacles or the density of already existing edges. Fig 1 schematically shows the time development of a growing network where edges branch several times, merge, and terminate.

## Implementation

We next describe the computational implementation of the theoretical framework for spatial network growth described in the previous sections. Key input parameters for the simulation are the spatial environment and the seed points (e.g. cells) that define from where the first edges start to grow. The spatial environment is provided in the form of 3D image volumes that contain information about the physical structure of the growth environment (e.g. tissue structure), as well as about signals that influence the growth process (e.g. growth factors). This grid-based representation of the environment in an otherwise continuous model, as illustrated in Fig 1, makes it possible to study network growth in real tissue environments as measured e.g. with confocal or light sheet microscopy without explicitly stating their geometry by including

**Table 2. Simulation parameters.**

| Parameter | Typical Value | Biological Meaning |
|---|---|---|
| Aperture | $\Omega$ = 25 deg | Growth cone mobility, self avoidance |
| Step size | $l \in \mathbb{R}, l \sim f(\mu, \sigma)$ | Growth cone velocity |
| Splitting angle | $\theta_{\text{bran}}$ = 75 deg, $\theta_{\text{bif}}$ = 75 deg | Angle at which cones bifurcate or new branches form |
| Search field | 3x3x3 cube | Range of interactions with the environment |
| Probabilities | $p_{\text{bran}}$, $p_{\text{bif}}$ = 2%, $p_{\text{term}}$, $p_{\text{reac}}$ = 0.1% | Probability to branch/bifurcate/terminate/reactivate |
| Memory | m = 5 | Directional persistence of growth cone movement |
| Seed parameters | $X, Y, Z, \phi, \theta$ | Seed positions and directions of growth processes |

Table 2: List of parameters of the implementation with typical values, and their meaning in a biological context.

them as scalar data on a 3D grid or "image" layer. Besides these image data and the seed locations and directions, the simulation requires a number of model parameters such as step length, aperture and degree of persistence as input, listed in Table 2. The aperture is the opening angle of the "field of view" of the growth cone. Smaller values limit the possible directions it can turn to, resulting in self-avoidance as described e.g. for neurons [44]. The angles for branching and bifurcations depend on the underlying biological mechanism, with large angles e.g. for vessel sprouting [46] or small angles for actin-crosslinker mediated branching of cell processes [47]. The corresponding probabilities for branching and bifurcation describe how likely these processes occur in a growing edge, which in biological systems can be both intrinsically determined (e.g. typical rates in osteocyte processes [14]) or externally controlled as in neurite outgrowth [2]. The parameter "memory" sets the number of previous steps that contribute to the next growth direction and determines the persistence of the direction. Biologically, persistent motion can arise either from physical constraints (e.g. persistence length of microtubules) or from the intrinsic persistence of the biological process, e.g. in cell migration. All parameters are set at the beginning of the simulation but can change for each growth cone as a function of time or external signals to capture the corresponding biological mechanism in the simulation.

The PDFs for the growth processes are derived from the image volumes by calculating gradients and the structure tensor (Fig 3). These image volumes can contain multiple channels to describe not only structure, but also scalar and vector fields to describe forces or signaling particles. The spatial growth environment can also be dynamic, either via external time-dependent changes or via changes introduced by the growing network itself. Notably, it is by this reciprocal feedback between the growing network and its spatial environment that the network architecture turns into an emergent property that cannot be encoded in the growth rules alone.

The growing edges are implemented as instances of a growth cone class with the properties listed in Table 3. These include both internal parameters as well as functions that define how external cues are translated into growth directions. All object instances work on internal memories, are able to create new objects, and can inherit their parameters. The workflow and interaction of the different memories are summarized in Fig 4.

Prior to deciding on the growth direction according to Eq 8 angular, the cone checks if there is a merge, branch, bifurcation or termination event. For merging, the cone at position $\vec{p}_c$ performs a collision detection with all edges $E_n$ of an ensemble $n \in \Omega_E$ that are within its search field of radius $R_c$. $\Omega_E$ includes all edges of a cluster specified through its members that can interact during $t$ growth steps. A new node is created at the $i$th positions $\vec{p}_{i,j}$ of the $j$th edge

## 3D Image Volumes of Complex Environment          Multilayer Simulation Grid

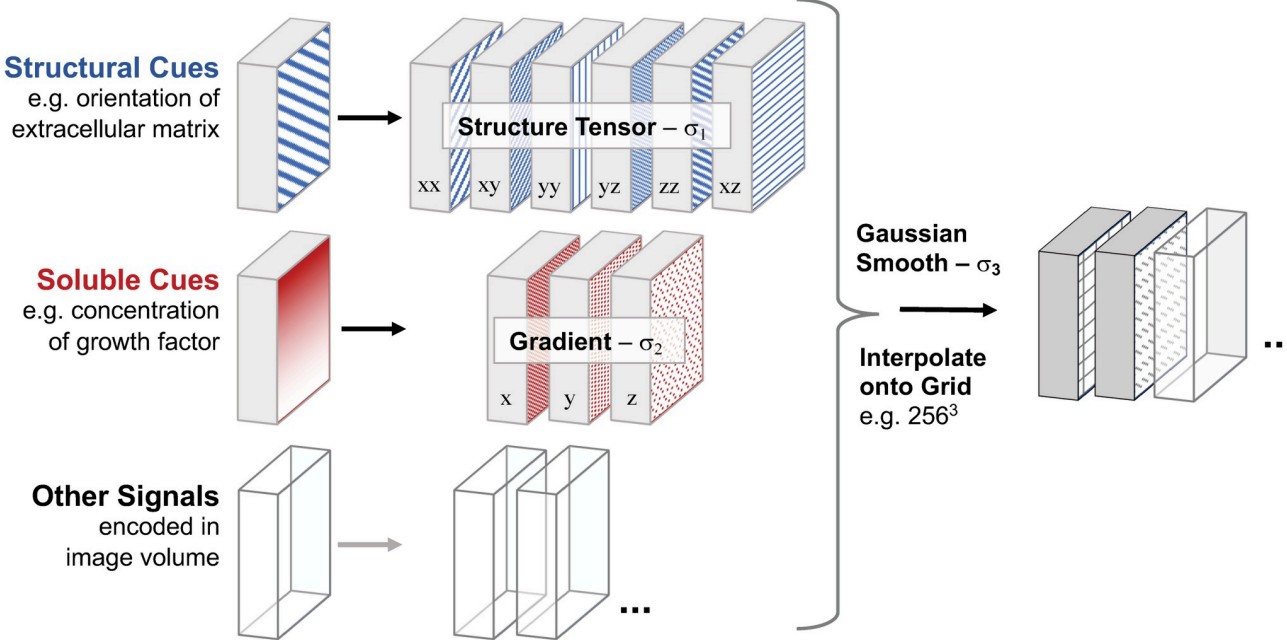

**Fig 3. Mapping the complex growth environment onto a multilayer simulation grid.** Structural and soluble cues and other signals are provided as 3D image volumes of real or simulated data (left). The six components of the structure tensor define the guidance cues from the substrate structure, whereas the three gradients of the images of soluble cues, e.g. growth factor concentrations, define the resulting directional bias. The feature sizes $\sigma_i$ used for the filters determine the resolution of the cues. Other signals can be integrated by computing the corresponding features (filters) that define the resulting growth cue. The image features containing the different cues are Gaussian smoothed and interpolated onto the final simulation grid. The growth simulation uses continuous coordinates on a discrete grid.

**Table 3. Global parameters, attributes and actions of model agents.**

| Global parameter | Type | Description |
|---|---|---|
| Instances | $i = 10$ | Number of global updates |
| Steps | $s = 25$ | Number of growth steps between updates |
| MGD Switches | Boolean | Presence of different interactions |
| **Agent property** | **Type** | **Description** |
| Positions | $[[x, y, z]_i, .., [x, y, z]_n]$ | History of all positions |
| Spherical angles | $[[\phi, \theta]_i, .., [\phi, \theta]_n]$ | History of all directions |
| First and last node | list[str] | Initial node and node of last merge/branch/bifurcation |
| Current state | str | 'growing', 'merged', 'terminated' or 'out of bounds' |
| **Action** | **Details** | |
| Growth | Construct $\text{MGD}_{ang}(\vec{\mu}, \Sigma)$, draw next direction | |
| Branch | Create new daughter cone at an angle, inherit from mother cone | |
| Bifurcate/Split | Create two new identical cones at an angle, delete old cone | |
| Merge | Merge current with nearby edge, create new node | |
| Terminate/Reactivate | Inactivate/reactivate cone, set status to 'terminated' or 'growing' | |

Table 3: Internal attributes of nodes and edges, data types, and description.

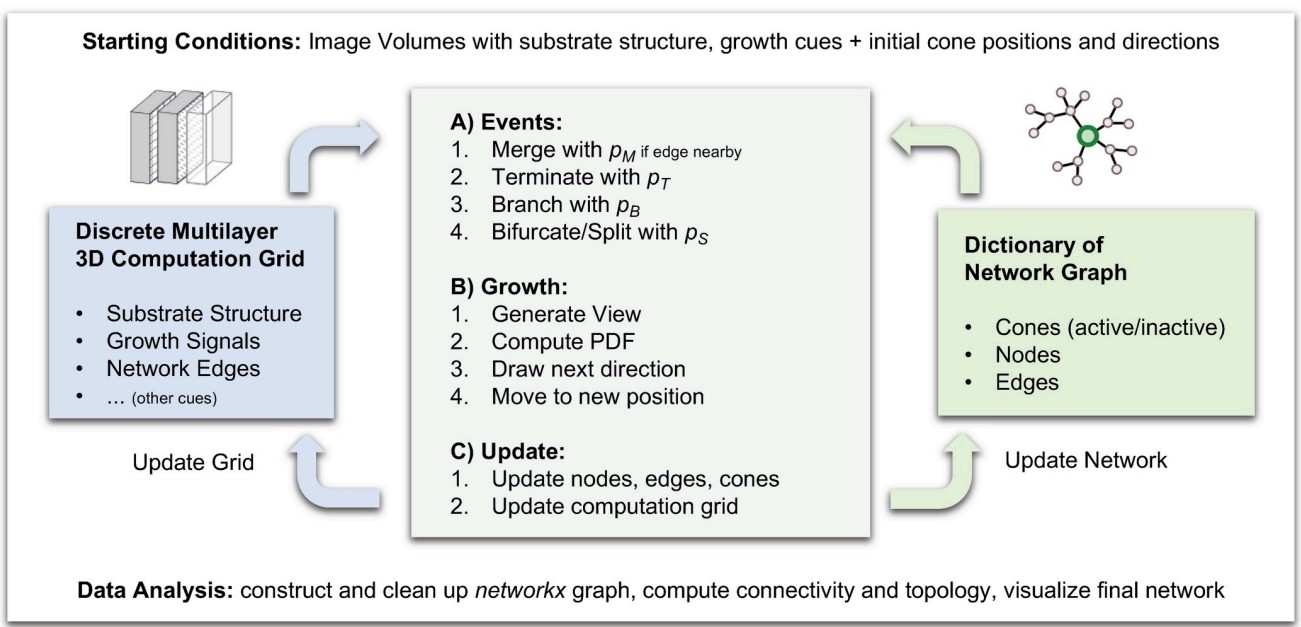

**Fig 4. Diagram of the simulation.** Starting conditions are provided as image volumes and as list of initial positions and directions. During each time step, the steps in the grey box are performed in parallel for each cone. The four elementary event types during network growth occur with their respective probabilities (A), depending on the local environment. Afterwards, the next position of each cone is determined from the spherical probability distribution computed from the local environment (B). Finally, the computation grid (left) and network dictionary (right), which are kept in shared memory, are updated by each parallel process, such that they can provide up-to-date conditions for all cones in the subsequent step. Finally, the network graph is constructed, its topological properties computed, and the network is visualized. These analysis steps can also be performed periodically during the simulation to monitor network development on-the-fly.

$j \in \Omega_E$ that fulfils the condition to be in shortest distance to the cone position.

$$\min |\vec{p}_c - \vec{p}_{i,j}|^2 \leq (2R_c)^2. \tag{11}$$

The cone terminates afterwards and the memory attributes of all participating objects are updated. Branching and bifurcation are executed via object functions. The events occur randomly with their respective probability. This probability and the parameters for branching angle and direction as well as the bifurcation plane are coupled to the external parameters of the spatial environment. The same type of coupling is implemented for termination and reactivation processes. If an agent branches or bifurcates, a new node is created at the splitting position. In the case of branching, a new cone object emerges from the new edge, and the mother branch updates its memory. For bifurcation, the old cone terminates, and two or more new cone objects are created and grow in new directions with an oblique angle to each other.

As the framework relies heavily on the communication of all objects, an efficient parallelization strategy is important. Already existing edges are not only stored in the graph dictionary, but also generate an imprint in an additional layer of the simulation grid. During growth, each process checks for the presence of existing edges by generating a view in this grid. If there is an edge in the vicinity, it can be looked up in the dictionary using its identifier from the simulation grid. To additionally reduce the number of collision calculations between actively moving cones, a Voronoi tessellation is performed for all growing edges to cluster neighbouring cones within a Delaunay distance $l_d$. Only those agents that are able to interact during the following $l_d$-steps have to communicate during these steps, while all clusters are independent from each

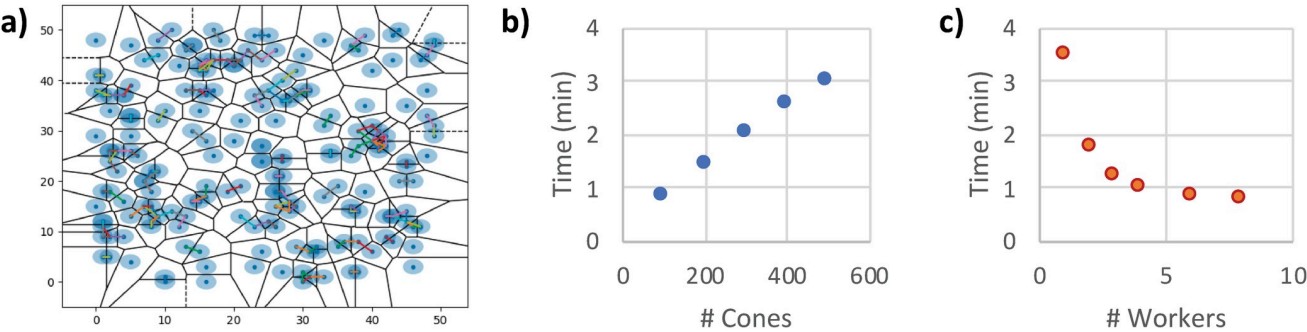

**Fig 5. Clustering and performance.** In (a), a 2-dimensional illustration of the clustering scheme to reduce the number of collision detections is shown for illustration. Blue dots correspond to active growth processes, and only processes with overlapping light-blue circles can interact. A Voronoi tessellation of the simulation volume is performed for all growing edges to cluster all neighbouring cones within a Delaunay distance $l_d$. Only agents that can interact during the following $l_d$-steps have to communicate during these steps, while all clusters are independent from each other and can be updated in parallel. In (b) and (c), the scaling of the simulation time with the number of growth cones and worker processes is shown.

other and can be updated in parallel. The clustering step is repeated after $l_d$ steps. In Fig 5, a 2-dimensional clustering is shown as an example. All agents with overlapping blue circles can interact and are part of the same cluster.

## Applications

We used our model to simulate the development of a multicellular biological network in a tissue with different degrees of anisotropy. This scenario is representative for a variety of biological systems where the final network architecture depends on the interplay of genetically encoded cell behaviour and the complex structured tissue environment. Examples include the osteocyte lacuno-canalicular system in bone [48], the development and regeneration of neuronal networks [2], or the reticular cell network in lymph nodes [15]. To show how our framework can be used to study the relationship between local growth rules, tissue structure, and global emergent network architecture, we performed growth simulations for different ranges of branching probability and angle, growth cone aperture, directional memory, as well as for different degrees of tissue anisotropy. To account for the stochastic nature of the growth process, we created a sample of ten random networks with identical starting parameters and calculated mean values. We then quantified the influence of the varying parameters on the resulting node number and average node degree, clustering coefficient, average shortest path length, edge density, as well as three types of node centralities. Closely spaced nodes are combined into a single node to enable comparison with experimentally obtained imaging data with finite spatial resolution [14]. Examples of the resulting networks and parameter relationships are shown in Fig 6, while the results of the sensitivity analysis of global network topology on the growth parameters are shown in Table 4. As can be seen in Fig 6A, the average node degree of the network gradually increases during the development of the network. A giant component emerges as all cells become connected to each other, indicating a critical point after around 30 steps. With increasing branching probability, the network becomes more dense, with higher mean degree and clustering coefficient and reduced average shortest path length (Fig 6B). Average node betweenness and information centrality decrease while harmonic centrality increases with higher branching rate. All three centralities approach the experimentally measured values in osteocyte networks [49]. We also find that stronger tissue anisotropy leads to a lower number of nodes but increasing clustering coefficient and average shortest path length,

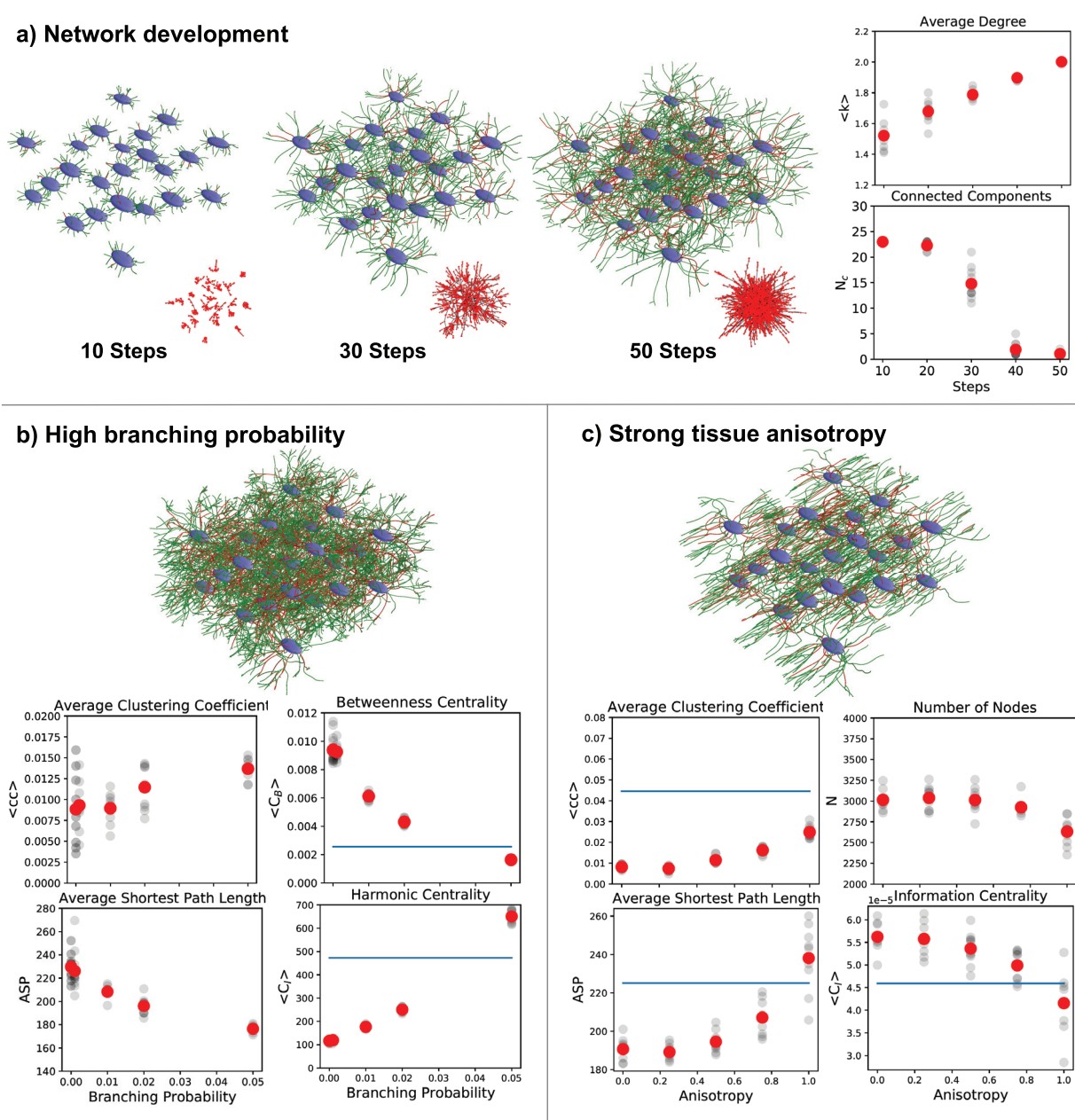

**Fig 6. Example simulations of multicellular network growth.** a) Cells (blue) emanate growth processes that form a dense, interconnected network of active (green) and terminated (red) edges. Node degree goes up (top) and a giant component emerges that connects all cells (bottom), as evident in the connectivity graphs (red/black). b) Higher branching probability leads to increasing clustering coefficient and shorter average paths. Node betweenness and harmonic centrality approach experimental values of osteocyte networks in bone (blue). c) Stronger tissue anisotropy leads to smaller networks with increased clustering coefficient and shortest path length, and decreased information centrality, approaching experimental values (blue).

while the average node degree remains unchanged (Fig 6C). These observations suggest that the ability of growing osteocyte processes to branch at a sufficiently high rate together with a sufficient degree of tissue organization are important parameters for the formation of viable osteocyte networks in bone. This example outlines the possibility for more systematic studies, e.g. to identify the underlying molecular and cellular parameters for observed changes in

**Table 4. Sensitivity analysis.**

| Parameter | $N$ | $\langle k \rangle$ | $\langle cc \rangle$ | ASP | $d$ | $\langle C_B \rangle$ | $\langle C_I \rangle$ | $\langle C_H \rangle$ |
|---|---|---|---|---|---|---|---|---|
| Directional Memory | ↗ | - | ↘ | ↘ | ↘ | ↘ | ↗ | ↗ |
| Growth Cone Aperture | - | - | ↗ | - | ↗ | ↗ | ↘ | ↘ |
| Branching Probability | ↗ | ↗ | ↗ | ↘ | ↘ | ↘ | ↘ | ↗ |
| Branching Angle | - | ↗ | ↗ | - | - | ↗ | - | ↘ |
| Tissue Anisotropy | ↘ | - | ↗ | ↗ | ↗ | ↗ | ↘ | ↘ |

Table 4: Sensitivity analysis of global network topology on local growth parameters and the anisotropy of the tissue environment. Columns from left to right: Number of nodes, average node degree, average clustering coefficient, average shortest path length, edge density, node betweenness, information and harmonic centrality.

network architecture in different biological systems. A similar comparison could be performed for any other 3D biological network for which such quantitative data are available. By modifying the provided code used to generate Fig 6 and Table 4, these examples can easily be adapted to other biological systems.

## Discussion

We developed a novel agent-based framework to simulate the growth of networks in three-dimensional space. The outgrowth of individual processes follows a 3D random walk with memory (correlation) and directional bias due to interactions with the environment. This framework can be used to simulate the growth of biological networks on all scales from single cells to ecosystems, or of generic network development through interaction of autonomous agents with each other and with external physical and chemical cues. During simulation, the network is directly converted into a graph using the networkx library [50], which provides access to many quantitative and comparative measures of network topology.

The first agent-based computer simulation of 2D spatial network growth, published in 1967 [20], already used a similar approach as our model: edges grow and branch according to rules, respond to attractive and repulsive cues, and follow density gradients. The next growth direction is randomly assigned based on sampling the space around the growth tip. Inspired by plant growth, this model can generate a variety of tree-like networks—a remarkable achievement given the limitations of computer simulations more than 50 years ago. While the field of complex networks flourished decades later, relatively little work was done on spatial network growth [1, 19]. In 2009, two new computational frameworks for neuronal network growth were published, CX3D [25] and NETMORPH [26]. Both use biologically realistic growing and branching rules and are openly available. In both models, synapse formation only takes place after the growth phase. CX3D offers a more physically realistic treatment of mechanical forces compared to our model, but lacks the flexibility of grid-based external structural cues. Two other neuron growth models [23, 51] efficiently generate neuron tree morphologies and space-filling networks, but without biologically realistic growth mechanisms and synapse formation. Taylor-King et al. [52] use an elegant mean field approach for network growth based on local state degree distributions. Whenever it is biologically plausible to analytically formulate local rules, such an approach can accurately reproduce global properties. In their model for vessel tree development, Perfahl et al. [53] include physical interactions with the environment and between edges to study the role of mechanical forces for vessel development. In their "Unifying theory of branching morphogenesis", Hannezo et al. [28] present a stochastic model for epithelial duct development based on branching and annihilating random walks. Although interactions are limited to explicitly defined chemical gradients and anisotropy, this model can

reproduce experimental tree topologies for a variety of organs. We summarized this comparison of our model with prior work in S1 Table. To our knowledge, our work presents the first framework for network growth that includes a biologically motivated local growth process as well as arbitrary interactions with an external environment, and is fully available as computational implementation.

We did not include energy consumption in our model in order to keep it as generic as possible. Depending on the type of network, different mechanisms of energy sources, energy distribution and consumption are possible. As an example, energy could either be "harvested" from the environment by the growing processes, or distributed throughout the network from cell bodies. Other existing models either omit energy consumption as well, or contain only specific energy-related mechanisms such as competition between growth cones [26] or diffusion of proteins from the soma [25].

One class of biological networks where the interaction and feedback between invidiual growth cones and the microenvironment during growth determines the resulting network architecture are multicellular networks, such as neuronal networks in the brain or the osteocyte network in bone. In bone, different degrees of tissue organization and the presence of soluble cues determines the connectivity and arrangement of the resulting network. With the framework presented here, the question can be investigated how local cellular response and resulting network topology depend on each other by conducting virtual experiments with varying parameters, which would not be possible in real experiments. The resulting theoretical understanding can then inform the design of guiding structures (scaffolds) to facilitate optimal network development in regenerating bone by providing the relevant physical and chemical signals [54].

The growth of vascular and neuronal cell networks during embryonal development is another example where soluble cues, the geometry of the environment, and the interaction between nearby cells all play an important role [55]. Recent improvements of lightsheet imaging technology now allow to visualize the entire growth process with high spatial and temporal resolution [56]. Our framework predicts not only the outcome, but also the dynamics of this process, which makes it possible to test specific hypotheses about the biological mechanisms that locally control this growth process. In the future, this might contribute to solving the question to what degree functional network architecture is encoded in the genome, or how the interactions and mechanochemical feedback loops between cells and the environment during growth determine the resulting tissue organization. Another interesting example is the role of curvature for tissue growth and organization [57]. With our framework, it will be possible to systematically vary the curvature of growth surfaces for networks while including other relevant biological and physical signals in the same simulation framework.

A very interesting related field is the exploratory growth of plants guided by pairwise interactions, tropic reponses to signals (e.g. light), and nastic (e.g. helical) movements. Recently, a computational model was introduced to model the dynamics of such sensory-growth systems [58, 59], taking into account the mechanical properties of the growing system. Such a framework offers interesting opportunities not only to understand biological control principles, but also for designing self-growing artificial systems. It would be interesting to see how this approach can be extended towards the development of functional network architectures.

While we developed our framework with network growth in mind, it can also be turned around and used to find and trace network structures in noisy image volumes by treating them as guiding signals for growth. Many approaches exist to trace filaments [60, 61], but they often have difficulties with branching structures. By simulating multiple growth processes either directly on image data or e.g. on probability maps predicted by convolutional neural networks, our framework might be able to find also highly branching and irregular structures.

Network growth as defined here is a parallel process by definition and thus can be computed in a distributed manner. We describe an efficient strategy for parallelization while taking into account the interaction between neighbouring growth cones. Future improvements could make use of the massively parallel processing power of graphical processing units.

## Conclusion

We developed a probabilistic agent-based model to describe individual growth processes as biased, correlated random motion with rules for branching, bifurcation, merging and termination. Using directional statistics, the influence of external cues on the growth process could be obtained directly from real image data. Our tool allows us to study the relationship between biological growth parameters and macroscopic function, and can generate plausible network graphs that take local feedback into account as basis for comparative studies using graph-based methods. Our approach is not limited to a specific type of network, includes a thorough treatment of probabilities, and can easily include arbitrary external constraints. We implemented our model in python and make it freely available as a computational framework, that allows other researchers to perform simulations for diverse types of biological networks on all scales.

## Methods

All edges and nodes are instances of their respective classes, with the attributes shown in Table 3. The framework was implemented in Python 3 using *numpy* [62, 63]. The network is directly converted into a networkx graph for further processing and analysis. On a 8-core Intel Xeon 6134 3.2 GHz machine with 64 GB RAM, running a network simulation with 100 growth cones in a 256x256x256 tissue volume for 50 steps takes 60 seconds. The scaling of the simulation time with the number of growth cones and worker processes is shown in Fig 5. Source code, documentation and example configurations of our computational model are available at https://github.com/CIA-CCTB/pythrahyper_net.

## Supporting information

**S1 Text. Detailed description of 3D random walks in complex environments.**
(PDF)

**S1 Table. Extended version of Table 1 with different external signals and cues.**
(PDF)

**S2 Table. Comparison of our framework with prior work.**
(PDF)

## Author Contributions

**Conceptualization:** Torsten Johann Paul, Philip Kollmannsberger.

**Data curation:** Philip Kollmannsberger.

**Formal analysis:** Torsten Johann Paul, Philip Kollmannsberger.

**Investigation:** Torsten Johann Paul, Philip Kollmannsberger.

**Methodology:** Torsten Johann Paul, Philip Kollmannsberger.

**Project administration:** Philip Kollmannsberger.

**Resources:** Torsten Johann Paul, Philip Kollmannsberger.

**Software:** Torsten Johann Paul, Philip Kollmannsberger.

**Supervision:** Philip Kollmannsberger.

**Validation:** Torsten Johann Paul, Philip Kollmannsberger.

**Visualization:** Torsten Johann Paul, Philip Kollmannsberger.

**Writing – original draft:** Torsten Johann Paul, Philip Kollmannsberger.

**Writing – review & editing:** Torsten Johann Paul, Philip Kollmannsberger.

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
