## [Decision Letter · Decision Letter 0]

27 Jul 2020

Dear Prof. Kollmannsberger,

Thank you very much for submitting your manuscript "Biological network growth in complex environments - a computational framework" for consideration at PLOS Computational Biology.

As with all papers reviewed by the journal, your manuscript was reviewed by members of the editorial board and by several independent reviewers. In light of the reviews (below this email), we would like to invite the resubmission of a significantly-revised version that takes into account the reviewers' comments.

Please note the attached reproducibility report, which should help you improve the reproducibility of your manuscript.

We cannot make any decision about publication until we have seen the revised manuscript and your response to the reviewers' comments. Your revised manuscript is also likely to be sent to reviewers for further evaluation.

Sincerely,

Pedro Mendes, PhD

Associate Editor

PLOS Computational Biology

Jason Papin

Editor-in-Chief

PLOS Computational Biology

Reviewer's Responses to Questions

**Comments to the Authors:**

Reviewer #1: Reproducibility Report has been uploaded as an attachment.

Reviewer #2: This manuscript addresses a major gap in our understanding of complex life: exploring generative processes which give rise to multicellular assemblies. It is very relevant work to an under-researched area.

An agent-based model is presented which gives rise to complex branching structures. The model is well described. The outputs of this generative process appear to be interesting and potentially relevant.

There are several areas which can be expanded upon to strengthen this manuscript:

- The biological relevance of the generative model could be made more explicit. In particular, how the various parameters in the model might be implemented or represented in a biological system. This would help support the inclusion of these parameters and being tangible modulators, rather than a series of variables capable of reproducing biological observations, but detached from how these processes actually occur.

- The networks generated by the model visually appear similar to the 2 biological networks they are compared to in the study. However, no quantitative comparison is provided. Using a variety of centralities to quantitatively compare the generative models to biological data would strengthen the conclusion that these models are capable of creating biologically meaningful outputs

- Following on from the previous point, a sensitivity analysis as to how the various parameters in the model impact the topological properties of the outputs from the model would be very useful. Again, using quantitative approaches via centrality measures, the range of topologies generated by the model, and the presence of critical points in parameters can be identified.

Reviewer #3: Major Comments

I enjoyed reading this paper, and I believe the premise of the paper - a generalized model for biological network growth under a variety of conditions - is extremely promising. The authors put great thought and effort into deriving the mathematical principles of network growth (in particular, I thought it was very clever to model individual environmental cues as multivariate Gaussians that can be combined through convolution into a single aggregate "environment" constraint). I am by no means an expert in directional statistics or geometry (and I have some points of confusion described below) but from what I understood, the mathematical foundations of their network growth model appear sound.

While the methods presented in this paper are exciting, I have concerns about publishing this without more empirical results. The authors present two anecdotal examples of using their network growth model to generate networks similar to those observed in biology. These two examples are not very convincing; in Figure 6, it is not obvious to me that the simulated network actually replicated the original network. And even if it did, these are just two examples. The authors are advertising a mathematical framework with a robust ability to generate a wide variety of biological networks by simply tweaking the input parameters, but they don't provide very much evidence of this. I realize this is easier said than done, but I think the paper would be strengthened with quantitative evidence that their network growth model can reproduce the growth seen in a variety of systems. This would likely involve finding a particular set of parameters that, when used in simulations, generates a sample of random networks that are then shown to be quantitatively similar to observed networks from the same system (and then repeating this process for other systems).

Minor Comments

- typo: "our tool allows to study..."

should this be "allows us to study..."? Note that this phrasing occurs throughout the paper, and if it is changed here it should be changed throughout.

- line 51-52: "making too many assumptions about these cues"

- line 270-271: "not modelled explicitly, e.g. in terms of hard-coded conditions, but through a generic mechanism based on angular probability density functions."

I think the authors would do well to more explicitly explain how their model for directional

cues differs from prior work.

- are equations (1) and (2) necessary? As the authors state, an analytic solution is elusive, and the authors end up resorting to a numerical solution based on random walks.

- I'm having some trouble understanding lines 82-90. In particular, how should I interpret the "mean vector" \\mu? Does this encode the location of some signal? Or does it encode the direction in which the current edge is being "pulled"? If it's the former, I think this needs to be better explained. If it's the latter, I believe it would be helpful to include a subscript, i.e. \\mu_t, to indicate that this is a time-dependent value.

This confusion carries on throughout the paper, and I think it could use a more intuitive explanation in lines 82-90.

- line 96 - should "determinate" be "determinant"?

- why does the document intermittently stop listing line numbers?

- I am slightly confused about the section on projecting onto the unit sphere. My intuition is that we essentially generate a random direction vector, and then project it onto a sphere of a randomly chosen radius? Is this correct? If so, I think the paper would benefit from explaining this explicitly.

- the MGD in equation (3) takes three inputs; but the MGD inside the integral in equation (7) takes only two inputs; why the disparity?

- line 162: should there be seed directions, in addition to seed locations?

- I could not understand the purpose of representing the environment as a grid. Is this simply for clustering to efficiently compute collision and merge events? Otherwise, I am not sure what difference this makes since the coordinates of edges are still continuous.

- Figure 6: are the blue/green square drawing, and the blue circular drawing, both graphical representations of the same simulation? This was confusing to me

Reviewer #4: The author developed a computational model of biological network growth in complex environment. A computational framework is based on directional statistics to model network formation in space and time under arbitrary spatial constraints. Growth is described as a biased correlated random walk where direction and branching depend on the local environmental conditions and constraints. To demonstrate the application of the computational model, authors performed growth simulations of the osteocyte lacuno-canalicular system in bone and of the zebrafish sensory nervous system.

It does a nice job of introducing relevance and background for such a question and overall I think it is interesting computational material. Unfortunately, a couple of things have put me a bit off enjoying the quality and impact of the paper's results:

1. The structure of extracellular space is named “images”. What is description of “images”? A more detailed description of the «image» concept must be given.

2. The variable p is introduced in the equation 1, but it is not written what this variable means in the framework of the model.

3. On page 5 (line 78) the vector u is indicated in bold, although throughout the entire article vectors are indicated by an arrow at the top. It is necessary to use the same designations throughout the article.

4. It is necessary to describe a possible biophysical mechanisms of “self-avoidance”, and references to experimental works must be provided.

5. What does means parameter “memory” in Table 2? There is no specific description or explanation in the text. Does this mean computational memory?

6. The model does not take into account energy consumption of growing branch at all. The authors should to explain in the text of the article, why this important process (which can create restrictions) does not included into the model.

7. For what purpose is table 4 in the article? This is purely a description of Python data types. I think this table needs to be dropped.

8. Authors used the model to simulate the development of two biological network systems: the osteocyte lacuno-canalicular system in bone and the sensory nervous system in Zebrafish. But they obtained only qualitative conclusions that states, that there is growth, and the images obtained are similar to experimental pictures. I believe that it is necessary to provide some quantitative characteristics that would allow readers to estimate how close the simulation results are to experimental results (for example, maybe by using the metrics of random graphs theory).

9. There are various models that describe growth in biological systems and networks. Authors should provide a brief comparison of their model with existing models, and should specifically indicate the advantages of their model over others.

10. In «Conclusions» (line 327) a reference to a specific paper(s) must be given.

**Have all data underlying the figures and results presented in the manuscript been provided?**

Reviewer #1: None

Reviewer #2: Yes

Reviewer #3: Yes

Reviewer #4: Yes

PLOS authors have the option to publish the peer review history of their article (what does this mean?). If published, this will include your full peer review and any attached files.

Reviewer #1: **Yes: **Anand K. Rampadarath

Reviewer #2: **Yes: **George Bassel

Reviewer #3: **Yes: **Arjun Chandrasekhar

Reviewer #4: No
---

## [Decision Letter · Decision Letter 1]

29 Oct 2020

Dear Prof. Kollmannsberger,

We are pleased to inform you that your manuscript 'Biological network growth in complex environments - a computational framework' has been provisionally accepted for publication in PLOS Computational Biology.

Best regards,

Pedro Mendes, PhD

Associate Editor

PLOS Computational Biology

Jason Papin

Editor-in-Chief

PLOS Computational Biology

Reviewer's Responses to Questions

**Comments to the Authors:**

Reviewer #2: The authors have addressed all concerns raised

Reviewer #3: No further comments

Reviewer #4: The authors have done a serious job. The article is now much better than the first version. I believe that the article can be accepted for publication.

**Have all data underlying the figures and results presented in the manuscript been provided?**

Reviewer #2: Yes

Reviewer #3: Yes

Reviewer #4: None

PLOS authors have the option to publish the peer review history of their article (what does this mean?). If published, this will include your full peer review and any attached files.

Reviewer #2: No

Reviewer #3: **Yes: **Arjun Chandrasekhar

Reviewer #4: No

---

## [Editor Report · Acceptance letter]

24 Nov 2020

PCOMPBIOL-D-20-00894R1 

Biological network growth in complex environments - a computational framework

Dear Dr Kollmannsberger,

I am pleased to inform you that your manuscript has been formally accepted for publication in PLOS Computational Biology. Your manuscript is now with our production department and you will be notified of the publication date in due course.

With kind regards,

Nicola Davies
